# Comparative Studies on Resampling Techniques in Machine Learning and Deep Learning Models for Drug-Target Interaction Prediction

**DOI:** 10.3390/molecules28041663

**Published:** 2023-02-09

**Authors:** Azwaar Khan Azlim Khan, Nurul Hashimah Ahamed Hassain Malim

**Affiliations:** School of Computer Sciences, Universiti Sains Malaysia, Pulau Pinang 11800, Malaysia

**Keywords:** drug-target interaction, data resampling, machine learning, deep learning, class imbalance, SMOTE

## Abstract

The prediction of drug-target interactions (DTIs) is a vital step in drug discovery. The success of machine learning and deep learning methods in accurately predicting DTIs plays a huge role in drug discovery. However, when dealing with learning algorithms, the datasets used are usually highly dimensional and extremely imbalanced. To solve this issue, the dataset must be resampled accordingly. In this paper, we have compared several data resampling techniques to overcome class imbalance in machine learning methods as well as to study the effectiveness of deep learning methods in overcoming class imbalance in DTI prediction in terms of binary classification using ten (10) cancer-related activity classes from BindingDB. It is found that the use of Random Undersampling (RUS) in predicting DTIs severely affects the performance of a model, especially when the dataset is highly imbalanced, thus, rendering RUS unreliable. It is also found that SVM-SMOTE can be used as a go-to resampling method when paired with the Random Forest and Gaussian Naïve Bayes classifiers, whereby a high F1 score is recorded for all activity classes that are severely and moderately imbalanced. Additionally, the deep learning method called Multilayer Perceptron recorded high F1 scores for all activity classes even when no resampling method was applied.

## 1. Introduction

Drug discovery plays an important role in the pharmaceutical and medical fields. In drug discovery, the prediction of drug-target interactions (DTIs) is the key to identifying potential drugs. Drugs in DTIs are usually chemical compounds, while targets are proteins [1]. Through the prediction of DTIs, high profits can be obtained, especially in the pharmaceutical field [1]. Hence, the demand to further identify potential interactions among drugs and targets has spiked an interest among researchers in pharmaceutical labs to perform DTI predictions [1]. DTI prediction can be done through several computational methods such as molecular docking and machine learning [1]. To perform these computational methods, the chemical compounds are usually represented as a Simplified Molecular Input Line Entry System (SMILES) string [2]. The SMILES notation is a user-friendly and easy-to-interpret notation often used by scientists to represent molecular structures of a chemical compound in computers [2]. Chemical compounds can also be represented as molecular fingerprints. Extended-Connectivity Fingerprints (ECFP) is the latest fingerprint methodology that is widely used in computational chemistry such as in drug-target interaction prediction, similarity searching and clustering [3].

Machine learning is currently one of the most significant and rapidly evolving topics in computer-aided drug discovery [4]. Machine learning is the preferred choice for DTI prediction as it enables large-scale testing of candidates within a short span of time, hence, making it easier for scientists and researchers to predict DTIs [1]. Machine learning can further be classified into supervised and unsupervised learning [4]. Supervised machine learning algorithms such as Naïve Bayes (NB), Random Forest (RF), Support Vector Machine (SVM) and *k*-Nearest Neighbour (kNN) are widely used in drug discovery, specifically in drug-target interaction prediction [4,5]. Unsupervised machine learning algorithms such as *k*-Means Clustering and Hierarchical Clustering can also be used for drug-target interaction prediction [4].

Furthermore, in 2013, Merck posted a multi-problem Quantitative Structure-Activity Relationship (QSAR) machine learning challenge [6]. The challenge was won by a deep learning network with a relative accuracy of approximately 14% over Merck’s in-house systems and even resulted in an article in The New York Times [6]. Since the challenge, advanced chemocentric machine learning methods with a focus on emerging deep learning technologies are being presented [6]. Deep learning architectures such as Convolutional Neural Networks (CNNs) and Deep Neural Networks (DNNs) appear to be well-suited for DTI prediction because they allow multitask learning and automatically construct complex features [6,7]. Deep learning is currently the most popular technique in drug discovery [8].

The accuracy of computational methods such as machine learning and deep learning plays a big part in determining whether a prediction was made successfully or not, and the predictive accuracy in any algorithm usually depends on the dataset that is being used. Including too many noisy variables in a dataset may reduce the accuracy of the prediction and lead to the over-fitting of data, which often produces promising but non-reproducible results [9]. Usually, real datasets cannot be directly fed into a learning algorithm due to class imbalance [10]. Class imbalance occurs when one class is represented by greatly more (majority) samples than another (minority) in binary classification [11]. When an imbalanced class is used, the classification of data may be negatively affected [10]. Machine learning algorithms are generally inclined by imbalanced data because most standard learning algorithms expect balanced class distribution [10]. Therefore, learning classification techniques are poorly achieved with imbalanced data [10,11]. Furthermore, according to [12], one of the major problems in DTI prediction is the existence of no true negative interactions and extreme class imbalance. These problems often affect the predictive performance of even powerful learning algorithms devastatingly [12]. 

According to [11,13], there are two possible ways to solve the problem of imbalanced classes, either by modifying the learning classification algorithms or modifying the data that is being presented to them. The authors in [11] have decided to modify the imbalanced data instead of modifying the learning algorithms because most machine learning algorithms are trained based on the assumption that the ratios of each class are equal. Furthermore, according to [14], one strategy to address class imbalance in learning algorithms is to generate one or more datasets, each with a different class distribution from the original dataset. Hence, two main categories of data resampling are utilized, undersampling and oversampling [13,14]. Undersampling involves the process of discarding instances from the majority class, while oversampling adds new instances to the minority class in order to achieve a balanced dataset, as discussed in [10,11,13,14].

One example of an oversampling strategy that can be applied is the Synthetic Minority Oversampling Technique (SMOTE) [13,14]. SMOTE oversamples the minority class in the dataset by synthesizing fake minority data into the original dataset, allowing the minority class to be balanced with the majority class [11,15]. Additionally, if new instances are added randomly to the minority class of the original dataset, the technique is called Random Oversampling (ROS) [13,14]. If the random discarding of instances from the majority class is performed, the process is known as Random Undersampling (RUS). Other resampling strategies include cross-validation and Adaptive Synthetic Resampling (ADASYN) [9,14,15]. Cross-validation is a resampling method where parts of the original dataset are sequentially left out and a multivariable analysis is conducted repeatedly until the entire sample has been assessed [16].

Nevertheless, the problems of imbalanced datasets in the binary classification of DTI prediction have not been properly solved yet. Class imbalance in general was addressed by various authors in other fields. For example, the authors in [10] performed an analysis of resampling an imbalanced class of a heart failure dataset, while in [11], the authors performed experiments to compare various resampling strategies on a clinical dataset. Furthermore, the authors of [15] have also performed an analysis on the in-silico prediction of blood-brain permeability of compounds using machine learning and resampling methods to overcome class imbalance. However, no comparative studies have been done on chemical datasets to determine the best resampling method to be used in the prediction of drug-target interactions. As mentioned earlier, the authors in [12] focused on the binary classification of multiple activity classes at once, instead of a single activity class. Moreover, the authors even proposed a new learning method where DTI prediction is addressed as a multi-output prediction task by learning ensembles of multi-output bi-clustering trees (eBICT) instead of using the available resampling techniques to balance their dataset, which leaves the problem of class imbalance in the binary classification of a single activity class in DTI prediction unanswered.

The effectiveness of deep learning algorithms in DTI prediction is also a popular topic of research in chemoinformatics. As previously mentioned, deep learning is currently the most successful technique in drug discovery and it has been proven to be successful in many other chemoinformatic fields, such as drug toxicity prediction and drug synergy prediction [8]. Moreover, in recent years, deep learning algorithms such as Artificial Neural Networks (ANNs), Convolutional Neural Networks (CNNs) and Recurrent Neural Networks (RNNs) have won numerous contests in pattern recognition and machine learning [17]. Deep learning algorithms are more advanced and complex and they are suited for drug-target interaction prediction because they allow multi-task learning as well as the extraction of complex features [7]. Furthermore, deep networks also provide hierarchical representations of a compound. However, the authors in [18] stated that in the existing methods to predict DTIs, this is generally treated as a binary classification problem which leads to severe class imbalance. Although the problem was addressed, the authors have instead decided to develop an ensemble classifier to overcome class imbalance by integrating several resampling techniques within the classifier itself [18]. Moreover, deep learning algorithms are also known to perform better than baseline machine learning algorithms and they are mostly used to represent complex features of a compound [7,19].

At the same time, [1,17] did not address the class imbalance problem in their datasets for DTI prediction, and yet both these works yielded high accuracy values without the use of any of the resampling methods mentioned by [13,14]. Hence, we would also like to take this opportunity to experiment with deep learning algorithms to study the effectiveness of deep learning methods without the use of any resampling method and compare its results with the machine learning algorithms using resampling techniques to overcome the class imbalance problem in DTI prediction. Furthermore, recent DTI problems are also becoming more advanced and they are now treated as a multiclass classification problem rather than binary classification [6,7,18]. Thus, in this study, we will address the class imbalance problem as a binary classification problem in which we will be focusing on a single activity class for both machine learning and deep learning methods in the hope of overcoming class imbalance in the datasets.

## 2. Related Works

### 2.1. Machine Learning Methods in DTI Prediction

The process of discovering and developing new drugs is extremely long and costly [20,21]. It can take up to 10 to 15 years of conducting research and testing to develop new drugs [21]. However, over the past decade, the field of artificial intelligence (AI) has moved from theoretical studies to real-world applications, especially in terms of drug discovery [22]. This is due to a number of factors such as the wide availability of new computer hardware, e.g., graphical processing units (GPU), the availability of big data of different sizes and types, powerful toolkits and the increase in computational capacity [21,22,23]. In this context, machine learning (ML) techniques have become extremely important in the pharmaceutical industry due to their ability to accelerate and automate the analysis of the large amount of data that is available in a short amount of time [21]. Hence, the use of machine learning methods has become a mainstream technique for analysing and solving problems involved in drug-target interaction prediction studies [23].

Several review papers discussing the use of machine learning techniques in drug discovery have been published in the past several years. In the following subsections, we will be discussing popular machine learning methods that are used in DTI prediction such as Support Vector Machines (SVM), Random Forest (RF), Naïve Bayes (NB), *k*-Nearest Neighbour (kNN) and Decision Trees (DT).

#### 2.1.1. Support Vector Machine

Support Vector Machine (SVM) is a popular classifier that is used in many fields, especially in drug discovery [5]. The objective of an SVM is to find a partitioning hyperplane in the sample space of the training set to separate samples of different categories [20]. In other words, SVM uses a nonlinear kernel function to map data into a high-dimensional space by searching for a separated hyperplane, where the hyperplane is then fitted in to maximize the margin between support vectors, pointing toward the nearest decision boundary and is then expressed as a linear combination of data points [4].

In the context of DTI prediction, an SVM constructs a hyperplane or a set of hyperplanes to predict the absence or presence of interactions between drugs and targets [4,24]. Since most existing studies on DTI prediction are treated as binary classification tasks, SVM is frequently used in predicting DTIs due to its significant accuracy and lower computational power requirements [23]. In [25], the authors used SVM to predict drug-target interactions and protein-chemical interactions at a genomic scale, whereby the method proposed relies on common chemoinformatics representation of the proteins and chemicals. The authors first predicted whether proteins can catalyse reactions that are not present in the training set [25]. Then, they proceeded to predict whether a given drug can bind with a target in the absence of the binding information of the drug and target. Furthermore, the authors in [25] also used a signature kernel and signature product kernels to predict drug-target interactions. The results showed that the authors were successful in predicting DTIs with the absence of binding information using SVM.

In [26], the authors also predicted drug-target interactions using SVM. They proposed a computational model of DTI prediction whereby molecular substructure fingerprints and Multivariate Mutual Information (MMI) of proteins and network topology are used to represent drugs and targets, as well as the relationship between them [26]. SVM was then used along with Feature Selection (FS) to build the classifier model to predict DTIs. The proposed model yielded good results where the values of area under the precision-recall curve (AUPR) showed an increase of 0.016 on the Ion Channel (IC) dataset that was used [26]. The authors further compared their models with other existing methods in which their model had the second-best performance on the Enzyme and GCPR datasets respectively.

#### 2.1.2. Naïve Bayes

Naïve Bayes classifiers are frequently used in chemoinformatics, especially in the field of drug discovery and DTI prediction [27]. Naïve Bayes (NB) algorithms are a subset of supervised learning methods that are essential in predictive modelling classification [28]. Bayesian methods are generally based on the Bayes theorem in which the formula can be mathematically derived to describe the probability of an event, as shown in the equation below (Equation (1)).
(1)$$P\left(A|B\right)=\frac{P\left(B|A\right)\;P\left(A\right)}{P\left(B\right)}$$

The equation above describes the probability *P* for state *A* existing for a given state *B.* In other words, Bayes used the probability of *B* existing given that *A* exists, multiplied by the probability that *A* exists, and normalized by the probability that *B* exists [27]. In the context of machine learning, Bayes classifiers assign the most likely class of each sample according to the description that is given by the vector values of its variables [21]. Furthermore, Bayesian classifiers are also increasingly being used in drug-target interaction prediction and drug discovery due to their versatility and robustness [27].

In [29], the authors proposed a Bayesian classifier model known as BANDIT or Bayesian Analysis to determine Drug Interaction Targets. The model is integrated with multiple data types such as drug efficacies, post-treatment transcriptional responses and reported adverse effects to predict DTIs [29]. For each data type, a similarity score was computed for all the known drug-target pairs and then the pairs are separated into those that share at least one known target and pairs with no known shared targets [29]. BANDIT then converts the individual similarity score of each pair into a distinct likelihood ratio, after which they are then further evaluated using the AUROC, ROC and Precision-Recall values. It is proven that BANDIT achieved an accuracy of up to 90% in predicting DTIs on over 2000 small molecules [29]. It was also concluded that BANDIT is an efficient approach to accelerate drug discovery due to its ability to correctly predict DTIs.

Additionally, in [30], the authors developed a web service called TargetNet for the prediction of potential DTIs. TargetNet is a server that can make real-time DTI predictions based only on molecular structures [30]. The authors used multiple Naïve Bayes classifiers along with various molecular fingerprints such as ECFP2, ECFP4, ECFP6 and MACCS to build a predictive model to predict DTIs [30]. When the user submits a molecule to TargetNet, the server will predict the activity of the user’s molecule across over 632 human protein targets, thus, generating a DTI profile that can be used as feature vectors of chemicals for wide application in drug discovery [30]. The results showed that the model built yielded AUC scores ranging from 75% to 100%, hence, proving it to be a success [30].

#### 2.1.3. Random Forest

Random Forest (RF) is another popular classifier that is extensively used in drug discovery, especially in the prediction of DTIs. RFs are widely used in solving any type of problem in the bioinformatic and chemoinformatic fields and they are chosen due to their performance, speed and generalizability [5,21]. A random forest consists of a collection of tree-like classifiers, e.g., decision trees in which they are independent of each other [20]. The output of an RF is determined by the voting results of the classifiers. Furthermore, RF is explicitly designed for large datasets with multiple features and it is commonly trained for large inputs and variables [28]. In drug discovery, RFs are mainly used for feature selection, classification, or regression [28].

In [31], the authors regarded the prediction of drug-target interactions as a binary classification problem and used RFs to predict DTIs. The authors used various combinations of features in building the RF model, such as tuning the *ntree* values from 100 to 1000 [31]. The input feature vector of the RF was concatenated by the drug network topology and protein network topology, respectively. Moreover, these topologies are proposed to characterize interaction pairs on the basis of the ‘guilt-by-association’ principle. In other words, the authors encoded the interaction information based on the assumption that a drug would target a protein when most of its neighbours have interacted with the protein in the network [31]. The results showed that the proposed RF model achieved an accuracy value of 92.53%, which is 10% higher than the existing method that this model was compared with [31].

Furthermore, in [32], the authors also predicted DTIs using Random Forest algorithms, as well as SVMs. In this paper, they proposed a systematic prediction model to predict multiple DTIs from chemical, genomic and pharmacological data. The authors used both RFs and SVMs in their experiments and the performance of both models was compared. In the study, the authors stated that Random Forests introduce two sources of randomness into the trees, random training sets and random input vectors [32]. Due to the randomness of RFs, the model performs really well and it is more robust against overfitting of data when compared to other learning methods such as SVM [32]. Nonetheless, the experiments performed yielded an average concordance of 82.83%, a sensitivity of 81.33% and a specificity of 93.62% on both models (RF and SVM), which is quite impressive in the prediction of DTIs [32].

#### 2.1.4. Decision Tree

Decision Tree (DT) is also another well-known machine learning classifier that is actively used in DTI predictions. DTs are commonly depicted as a tree, with the roots on top and the leaves at the bottom [27]. The tree can then be split into two or more branches, and each branch may also further split into other smaller branches. In the context of machine learning, the trees and leaves are referred to as nodes, and the split of a branch is known as an internal node [27]. Additionally, DTs are also simple to understand as they are also easy to interpret and validate, hence, the reason it is chosen as a classifier in predicting DTIs.

The authors of [33] proposed an ensemble learning method called EnsemDT, which comprises an ensemble of DT classifiers to predict drug-target interactions. The drug features are extracted from the SMILES representation of the drugs from the database [33]. The target features are then extracted from the PROFEAT server and the drug-target pairs are represented as feature vectors [33]. To overcome the overfitting of data in the dataset, a resampling modification was embedded in the model to improve the predictive accuracy. The proposed model is then compared with other machine learning models such as the classic single Decision Tree (DT), Random Forest (RF) and Support Vector Machines (SVMs), and the results showed that the AUC value of EnsemDT was 0.906 overall, proving that the use of DTs in DTI prediction has positively impacted the AUC value of the ensemble method [33].

### 2.2. Deep Learning Methods in DTI Prediction

Deep learning is a subfield of machine learning that uses different Artificial Neural Networks (ANNs) with many layers of non-linear processing units for the purpose of data representations and to model high-level abstractions of data [34,35]. The concept of deep learning is often related to ANNs in principle, as the basic principle of ANNs and deep learning is the learning of layered concepts [5]. In deep learning architectures, each layer will train on a set of distinct features based on the output of the previous layers, depending on how many layers are present in the network [35,36]. The more layers, the deeper the neural network [36]. A deeper network means that more complex features can be learned by the nodes present in the network [35]. Moreover, deep learning networks are also capable of handling very large and high-dimensional datasets with millions of parameters, which is the reason why deep learning is very successful in a wide range of applications, especially in drug discovery and chemoinformatics [6,7,35]. Currently, there are many popular deep learning algorithms that are actively developed and used for the prediction of drug-target interactions (DTIs) such as Convolutional Neural Networks (CNNs) and Multilayer Perceptrons (MLPs).

#### 2.2.1. Convolutional Neural Networks

CNN or Convolutional Neural Network is a popular deep learning method that is frequently used in DTI prediction and drug discovery. CNNs are neural networks that act in grid-like structures and they are mainly used in processing images [37,38]. A CNN consists of several convolutional and pooling layers that are placed in an order that may be optional [13]. The convolutional layers learn a set of filters that extracts a set of local patterns (sub-features) in a local receptive field of the input, where the input can either be one-dimensional, two-dimensional, or even three-dimensional [37,38]. The pooling layers on the other hand enlarge the local receptive field by down-sampling the input of the layer [37].

In the context of DTI prediction, the authors of [39] proposed a method called DeepACTION to predict potential or unknown DTIs. DeepACTION is a deep learning-based method that comprises CNNs to accurately predict novel DTIs [39]. The model was integrated with MIMB (Majority and Minority Instance Balancing), which is a method that was developed by the authors to balance the dataset that is being fed into the neural networks [39]. The method was proven to be successful in predicting unknown DTIs when compared to other existing methods such as kNN and Naïve Bayes, achieving a stunning AUC value of 0.9836 (98.36%) [39].

In [40], the authors have also proposed a CNN-based approach for predicting DTIs. The proposed method is called DeepConv-DTI, a novel DTI prediction model that extracts the local residue of protein sequences using a CNN-based approach to accurately predict drug-target interactions [40]. As a result, DeepConv-DTI was proven to perform better than other deep learning models that were developed for the large-scale prediction of DTIs in terms of accuracy and F1 scores [40].

#### 2.2.2. Multilayer Perceptrons

Multilayer Perceptrons (MLPs), also known as feedforward neural networks, are one of the most popular types of Artificial Neural Networks (ANNs) [41]. MLPs provide an output based on a set of input sources and, in training MLPs, backpropagation is utilized. Multilayer Perceptrons are similar to directed graphs such that the input nodes consist of multiple hidden layers and the output nodes have some weight attached to them [41]. MLPs are generally very easy to use, which is why they are widely used in drug discovery, especially in drug-target interaction predictions. For example, [42] utilized MLPs in their proposed model, DTI-GAT. Two layers of MLPs are implemented in the feature encoder for protein and drug respectively and three other MLP layers are fused in the final interaction decoder with an input dimension of 512 neurons [42]. DTI-GAT was trained using the Adam optimizer and the proposed method achieved an accuracy of 72.90% and a precision value of 75.54% when compared to other deep learning models such as DeepConv-DTI and DeepDTA [42].

Furthermore, in [43] the authors proposed a method called UnbiasedDTI to predict DTIs. UnbiasedDTI is a deep ensemble-balanced learning method consisting of three main modules [43]. These modules are the drug encoder module, the protein target encoder module and, finally, the prediction module whereby each of the modules consists of MLPs of four linear layers with 1024, 256, 64 and 256 neurons [43]. With UnbiasedDTI, the authors found that their proposed model achieved the highest performance (83.8% of F1 score and a recall of 90.3%) when compared to other unbalanced deep learning models [43].

## 3. Results and Discussion

### 3.1. Machine Learning vs Machine Learning with Sampling

After conducting all the experiments with various machine learning approaches with the presence and absence of resampling techniques, the performance of the machine learning models was assessed using the accuracy, precision and recall metrics for each activity class and the detailed results are thoroughly explained in the Appendix A. Based on all the results discussed in terms of accuracy, precision and recall, we can say that the accuracy, precision and recall metrics are not enough to fully evaluate a model. The performance of the models highly depends on the dataset that is being fed to them for learning and predicting, regardless of whether it is balanced or unbalanced.

In this study, the dataset that we have used is highly imbalanced with the number of inactive compounds almost two times more than the number of active compounds for each of the activity class. It is also noticeable from the results different activity classes perform better and worse when different resampling methods are being used. This could be due to the difference in the number of active compounds per activity class. According to [44], the classification of the degree of imbalance for imbalanced data is as shown in Table 1.

To further investigate and determine the best machine learning classifier and resampling method combination, the results are sorted by the number of active compounds per activity class and the respective percentage of minority class for each cancer-related activity class (Table 2), which can be computed using (Equation (2)):(2)$$Percentage\;of\;Minority=\left(\frac{Total\;Number\;of\;Minority\;Instances}{Total\;Number\;of\;Instances}\right)\times 100\%$$

In general, Random Undersampling (RUS) performs the worst among all the resampling methods when applied to any of the classifiers developed, whereby a drop is always observed in terms of the precision and recall of each classifier model. However, RUS seemed to perform quite well in terms of precision and recall on the RF classifier in terms of the activity class, PDGFR-B (Appendix A). Note that PDGFR-B has only 41 active compounds (Table 2) and when RUS is implemented, the number of inactive compounds was severely undersampled, which will result in lost data; hence, although good recall and precision are recorded, the model is unreliable as most of the information about the activity class is lost due to severe undersampling.

Furthermore, SMOTE, ADAYSN, BorderlineSMOTE, SVM-SMOTE and SMOTETomek performed well with the RF classifier across all activity classes in terms of accuracy. Nonetheless, there is not one specific resampling method that stands out as the best, as different activity classes perform better with different resampling techniques. However, it is also important to understand that high accuracy does not mean that the model is performing predictions correctly, the precision and recall values also play an important role in evaluating a model’s performance, whereby high precision and recall values mean that the model is able to make correct predictions efficiently. Hence, in Table 3, it is found that the best resampling method varies across all activity classes when evaluated under different metrics. For example, in terms of precision, when no resampling is applied with RF across the activity classes BRAF, CDK-6, HER2, PDGFR-A and VEGFR-1, the precision value is the best compared to the other activity classes where resampling is required. It is also interesting to note that for activity classes that are severely and moderately imbalanced; MEK1, PDGFR-B, KRAS and PD-1 (Table 2), the precision is at its best when resampling methods are applied to the RF classifiers as depicted in Table 3.

In terms of recall, the results differ from the accuracy and precision values. Here, the combination of the GNB classifier with SVM-SMOTE performs the best for the activity classes BRAF, CDK-6, HER2 and PD-1, while other activity classes perform the best with SMOTE (KRAS), BorderlineSMOTE (PDGFR-A) and SMOTETomek (MEK1). Nonetheless, from Table 3 we can observe that there is no single resampling method that stands out as the best across all activity classes. Nevertheless, after analysing the results for all the classifiers with different resampling methods across all activity classes, it is obvious that the RF classifier is the best classifier in terms of its average accuracy, precision and recall, and for each activity class, the best resampling method differs as well. To visualize this, the average performance of the classifiers was computed as shown in Figure 1.

Based on Figure 1, the RF classifier performs the best in terms of accuracy, precision and recall, with an average accuracy of 96.19%, an average precision of 86.26% and an average recall of 83.29%. In conclusion, RF is the best classifier in general, and GNB is the second-best classifier among the other classifiers in terms of precision and recall, while SVM and DT are considered the weakest classifiers due to their low accuracy, precision and recall recorded throughout all the experiments conducted.

To further determine the best resampling method for each activity class with its pairing classifier, further evaluation based on the recall and precision values was done in order to compute the F1 score. Since the results obtained so far are not enough for us to determine the best overall resampling method, the F1 score was computed (Appendix A). Based on the F1 scores, we have managed to determine the best machine learning classifier and resampling method pair for each cancer-related activity class. The results, along with the respective F1 scores are shown in Table 4.

However, there are a few exceptions that we would like to highlight, especially for the PDGFR-B and VEGFR-1 activity classes. It is observed that when the RF classifier is applied with RUS for PDGFR-B, the F1 score is the highest with a value of 93.50%. As discussed earlier, RUS is an unreliable resampling method as it severely undersamples the inactive data, which might result in a huge loss in data. Thus, we have decided to choose the next highest F1 score value for PDGFR-B, namely, 85.68% using the GNB classifier with SVM-SMOTE as the resampling method.

Furthermore, for VEGFR-1, the pairing is denoted as RF + None, which indicates that the F1 score is the highest when no resampling method is used with the RF classifier, while with other resampling methods, a huge drop in the F1 score can be observed (Appendix A). To summarize, it is found that no specific resampling method performs well across all activity classes in the dataset. However, it was observed that the SVM-SMOTE resampling method showed promising results when paired with the RF and GNB classifiers on severely and moderately imbalanced activity classes such as MEK1, PDGFR-A, PDGFR-B, HER2 and KRAS.

### 3.2. Deep Learning with No Resampling

#### 3.2.1. Convolutional Neural Network (CNN)

Generally, CNN performs well on all activity classes in terms of accuracy, with values ranging from 93.89% to 99.76%, which in general is better than all the ML classifiers even when no resampling methods are applied (Refer to Figure 2). However, a drop in the precision values is observed among all activity classes, especially for activity classes that are severely imbalanced—MEK1 and PDGFR-B (Table 2). A huge drop in both the precision and recall values is also observed in the PDGFR-A and VEGFR-1 activity classes.

Additionally, MEK1 also records a 100% recall value despite a drop in the precision value, which means that CNN was able to find all the true positives (positive interactions between the drug and target).

#### 3.2.2. Multilayer Perceptron (MLP)

From Figure 3 we can observe that MLP in general performs well in terms of accuracy and F1 score. We have also found that when activity classes that are severely and moderately imbalanced, mainly CDK-6, KRAS, MEK1, and PDGFR-B (Table 2) were fed through the layers in MLP, the precision value recorded for all of them was 100% (Figure 3). Furthermore, with activity classes that are severely imbalanced (CDK-6 and PDGFR-B), a 100% accuracy, precision, recall and F1 score was recorded.

Both deep learning methods, CNN and MLP, performed exceptionally well in terms of accuracy across all the activity classes even when no resampling methods were applied. When comparing the performance of CNN and MLP, MLP on average was able to correctly predict positive interactions between a drug and target, especially for activity classes that are severely and moderately imbalanced. The F1 scores of MLP across all activity classes are also better than CNN. A further comparison in terms of the F1 score between the machine learning classifiers with their respective pairing with resampling methods and with MLP was done to study the effectiveness of deep learning methods in overcoming class imbalance in the binary classification of DTI prediction. The results of the comparison are summarized in Table 5.

From Table 5 we can observe that MLP performs better for almost all activity classes, except for the severely imbalanced activity class, PD-1 and the moderately imbalanced activity classes, PDGFR-A and VEGFR-1, whereby the F1 score is high for PD-1 when ADASYN is used with the RF classifier and when SVM-SMOTE is used with the GNB classifier for PDGFR-A. It is also interesting to note that for VEGFR-1, which is moderately imbalanced, no resampling is required when the RF classifier is used compared to MLP, where there was also no resampling method. Nonetheless, on average, MLP, the deep learning method where no resampling was applied, performed better than machine learning classifiers paired with various resampling methods with an average F1 score of 92.36%.

## 4. Materials and Methods

To overall methodology of this study is visualized in Figure 4.

### 4.1. Data Acquisition

The data used in this study were obtained from BindingDB. The BindingDB database is a public, web-accessible database consisting of over 2 million binding data for over 8816 target proteins and 1 million small molecules [45]. However, in this paper, we have only selected 10 activity classes to demonstrate DTI prediction in terms of binary classification to minimize the scope of our search, which is the target proteins in cancers. The activity classes selected are popular proteins that are used to detect and treat cancer in the human body. There are many target proteins that play a huge role in detecting and treating various types of cancer such as the HER2 protein for breast cancer and the BRAF protein in lung cancer [46,47,48]. The selected activity classes are listed in Table 6. Table 6 also highlights the number of active compounds that are interacting with each respective activity class along with the class’s abbreviation.

### 4.2. Data Preprocessing

To prepare the selected activity classes for the prediction of drug-target interactions (DTIs), the SMILES notation of each active compound for each activity class in the BindingDB dataset will be converted into chemical fingerprints. Before converting the SMILES notations into chemical fingerprints, we will first perform SMILES canonicalization. The SMILES notation or Simplified Molecular Input Line Entry System is the most popular annotation used by scientists to represent a chemical compound [2]. In drug discovery, it is common that a structure may be represented as many different SMILES strings, as one can start with any atom in a molecule to derive a SMILES string. Hence, it is important to represent each chemical compound with a unique set of strings, which is where SMILES canonicalization comes in. This process is done to eliminate redundancy in the SMILES string representation of the chemical compounds. Redundancy happens when compounds that are similar form a different conformation that affects the SMILES notation. To perform SMILES canonicalization, we have used an open-source chemoinformatic extension called RDKit within KNIME, a popular data analytics software used in drug discovery [49,50]. Hence, the SMILES notation of each active compound for each of the activity classes’ files will be used as an input to convert them into canonical SMILES.

The fingerprint representation that we will be using is the Extended-Connectivity Fingerprints (ECFP) representation. The idea of ECFP is to encode the structure of a molecule in a bit string of ‘1′ to represent the presence, and ‘0′ the absence, of a particular substructure in the molecule [3]. In this paper, the canonical SMILES notations were converted to ECFP fingerprints of radius 4 or ECFP4. ECFP4 is widely used in the field of chemoinformatics, specifically in DTI prediction as demonstrated in [51,52]. To perform the conversion from canonical SMILES to ECFP4, we will also use KNIME. The conversion can be done using the Chemistry Development Kit (CDK) extension within KNIME. CDK is an open-source library that is used in chemoinformatics, specifically in drug discovery [53]. CDK fingerprints were chosen over the fingerprints in RDKit due to a compatibility issue with KNIME in which, when RDKit was used in converting canonical SMILES to ECFP4, KNIME issued multiple warnings and promptly crashes even before the conversion process was complete. Thus, the CDK extension was chosen as our next alternative to perform the conversion efficiently. 

The first step in the conversion from SMILES to ECFP4 was to read all the files containing the SMILES notation (for each activity class) and then, using the notation as an input, convert it to canonical SMILES. Then, the canonical SMILES were converted into 1024 binary bits of ECFP4 fingerprints. Finally, the fingerprints were written into a new CSV file for further processing. The overall process is shown in the KNIME workflow in Figure 5.

After successfully converting from SMILES to ECFP4, the inactive data for each of the activity classes are generated through a dataset generation program, whereby, if the compound is active against a specified activity class, the target is set to ‘1′ and if it is found to be inactive, the target column will be set to ‘0′. The general idea of this program is to compare the active compounds of a specified activity class against all the other activity classes in order to mark the active and inactive compounds, whereby, if a match is found, then it is denoted as ‘1′ and if no match was found, then it is denoted as ‘0′. This process was repeated ten times for each activity class. Finally, to properly clean the data for further use, the duplicates and missing data were also removed. The final numbers of active and inactive compounds for each activity class at the end of this phase are listed in Table 7.

### 4.3. Predictive Modeling

The next step was to develop baseline machine learning models with the implementation of various resampling techniques. In this study, we have developed four baseline machine learning methods infused with six different data resampling techniques using the Python programming language with the help of the scikit-learn and imbalanced-learn libraries, as well as two baseline deep learning methods without the use of any resampling techniques.

#### 4.3.1. Machine Learning vs Machine Learning with Resampling

The four machine learning methods that are developed are Support Vector Machine (SVM), Random Forest (RF), Gaussian Naïve Bayes (GNB) and Decision Tree (DT), and the six resampling methods that will be used are Synthetic Minority Oversampling Technique (SMOTE), Random Undersampling (RUS), Adaptive Synthetic (ADASYN), BorderlineSMOTE, SVM-SMOTE and SMOTETomek. A brief explanation of each resampling method is given in Table 8.

Before applying any of the resampling techniques mentioned above within the machine learning models, the models will be optimized by tuning the parameters. Hyperparameter tuning is done so that the model with the best parameters is determined before feeding any data into it for learning. The parameters are defined based on the documentation in scikit-learn for each of the machine learning classifiers to be tuned [60]. For the RF classifier, the parameters that were considered for tuning are the number of estimators (*n_estimators),* the minimum number of samples required to split a node (*min_samples_split*), the minimum number of samples needed to be a lead node (*min_samples_leaf*), the maximum depth of the tree (*max_depth*) and the function to measure the quality of a split (*criterion*). For the GNB classifier, the parameter that was tuned is the *var_smoothing* parameter, which is a user-defined variable that is added to the default value of distribution variance, which is derived from the training dataset in order to smoothen the Gaussian curve in making predictions [60]. For the SVM classifier, the parameters considered for tuning are the *C* value, which is a regularization parameter, the *kernel* used (this variable helps the SVM to achieve the right mapping function to perform predictions) and finally, the *gamma* value which is the kernel coefficient that defines the influence of the training points in the dataset in order to perform predictions [61]. To tune the DT classifier, the parameters that were defined are similar to those in the RF classifier, with the addition of the *max_features* variable and the *ccp_alpha* values in which *max_features* is the maximum number of features considered to find the best split and *ccp_alpha* or Cost Complexity Pruning Alpha is a variable that can be used to control the number of nodes pruned in the tree to make predictions [61]. 

Hyperparameter tuning of the parameters defined above was done using the GridSearchCV function in scikit-learn [60]. To use GridSearchCV, we first need to create a dictionary of the parameters that we want to tune (as discussed above), and pass it into the function with the associated machine learning classifier. Then, the model will be fitted with the *X* (features, or ECFP4 fingerprints) and *Y* (target activity) data for the function to iterate and check for all combinations of all the parameters. The dataset used to perform hyperparameter tuning is a dummy dataset of our BindingDB dataset containing a random number of active and inactive compounds. At the end of this process, the best parameters are printed out and they are summarized in Table 9.

After tuning and determining the best estimator for all the machine learning models as depicted in the table above, the classifiers were infused with *k*-fold cross-validation where *k* is 10. The 10-fold cross-validation was applied to make sure that all the data are being tested. In this approach, the dataset is separated into ten (10) different folds. In each iteration of the code, one-fold is set aside for testing while the rest are used for training. This was done 10 times repeatedly until all the folds have been used for testing and training.

#### 4.3.2. Deep Learning with No Sampling

The two (2) deep learning models developed are Convolutional Neural Network (CNN) and Multilayer Perceptron (MLP). The deep learning models are developed with the help of the TensorFlow library for Python programming with no resampling technique applied. The architecture of the CNN that was developed is similar to the CNN found in [51], with the addition of two Batch Normalization layers. A screenshot of the model’s architecture is shown in Figure 6 below.

The architecture of MLP is similar to the one developed in [43], whereby, instead of four linear layers, three linear layers of size 256, 128 and 64 neurons were developed for training and testing. Both these deep learning models were infused with 10-fold cross-validation and will be fitted and trained across 10 epochs with a batch size of 256, and this process was repeated 10 times to accommodate each activity class.

### 4.4. Performance Evaluation

To evaluate the performance of each of the machine learning and deep learning models with or without resampling strategies, we have used the scikit-learn library to compute the performance metrics of the models, mainly the Accuracy, Precision, Recall and F1 values. The formula for accuracy is (Equation (3)):(3)$$Accuracy=\frac{\left(TP+TN\right)}{\left(TP+TN+FP+FN\right)}$$
where *TP* means True Positive (the number of drug-target pairs predicted as interactions correctly), *TN* stands for True Negative (the number of negative pairs predicted as non-interactions correctly) [62], *FP* means False Positive (the number of negative drug-target pairs classified as interactions incorrectly, and *FN* means False Negative (the number of positive drug-target pairs classified as non-interactions incorrectly) [62]. Hence, the accuracy value of a model means the number of correct predictions of interactions over the total number of predictions. The formula for the precision value is (Equation (4)):(4)$$Precision=\frac{TP}{TP+FP}$$

Thus, the precision value indicates how good the model is at predicting positive interactions between a drug (compound) and a target. Conversely, the recall value measures the model’s ability to detect positive samples, which in this case means detecting positive interactions between a drug and a target. The formula to calculate the recall value is (Equation (5)):(5)$$Recall=\frac{TP}{TP+FN}$$

The *F1* score is defined as the harmonic mean of the precision and recall values and this metric is essential to compare the performance of the models when a resampling method is or is not present. *F1* can be computed using (Equation (6)):(6)$$F1=2\times \left(\frac{Precision\times Recall}{Precision+Recall}\right)$$

A comparison and analysis will be made using the results obtained from the experiments of drug-target interaction (DTI) prediction using various resampling methods in machine learning models as well as DTI prediction in deep learning models without the use of any resampling methods for all 10 cancer-related activity classes selected.

## 5. Conclusions

In this study, machine learning and deep learning approaches were used to perform drug-target interactions on an imbalanced dataset by comparing different resampling techniques, namely, Synthetic Minority Oversampling Technique (SMOTE), Random Undersampling (RUS), Adaptive Synthetic (ADASYN), BorderlineSMOTE, SVM-SMOTE and SMOTE with Tomek Links (SMOTETomek). The imbalanced dataset consists of 10 different activity classes, all target proteins in cancer. The data collected in this study can be used as a benchmark dataset in order to predict drug-target interactions (DTIs) in cancer, especially in identifying and discovering new anticancer drugs in the near future. It was found that the use of Random Undersampling (RUS) in predicting drug-target interactions severely affects the performance of a model, especially when the dataset is highly imbalanced. Although high recall and F1 scores were observed for severely imbalanced activity classes, RUS is considered unreliable. This is due to the fact that, in drug-target interaction prediction, the active and inactive data within a dataset is extremely crucial in identifying new drug-target pairs. Hence, using RUS may be misleading since most of the data will be lost due to undersampling, thus, rendering it an unreliable resampling method for DTI predictions. Conversely, SVM-SMOTE can be used as a go-to resampling method when dealing with imbalanced datasets, especially when it is paired with the Random Forest (FR) and Gaussian Naïve Bayes (GNB) classifiers. With SVM-SMOTE, a consistently high F1 score was recorded for almost all activity classes that are severely and moderately imbalanced (over 85%). Last but not least, it is also important to note that the deep learning method, Multilayer Perceptron (MLP) recorded a constantly high F1 score of over 90% across all activity classes even when no resampling method was applied for DTI prediction. However, there is still room for additional resampling methods as well as their use with other deep learning and hybrid algorithms in predicting DTIs in cancer.

## Figures and Tables

**Figure 1 molecules-28-01663-f001:**
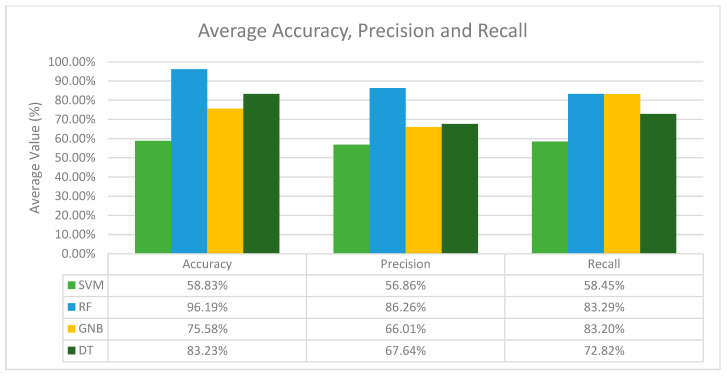
Average accuracy, precision and recall of all machine learning classifiers in general.

**Figure 2 molecules-28-01663-f002:**
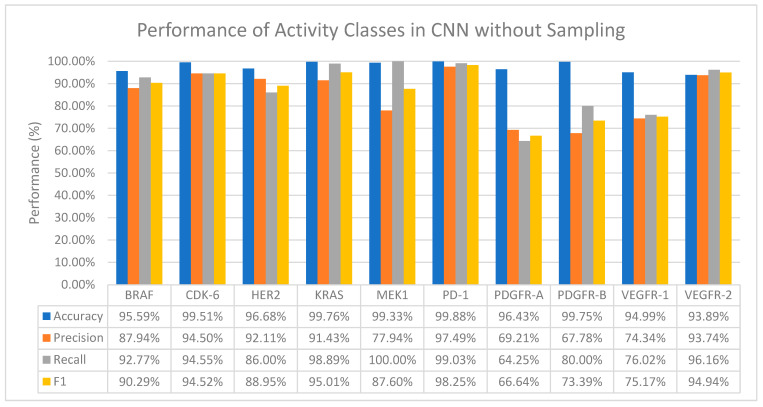
The performance of CNN in terms of accuracy, precision, recall and F1 score without sampling.

**Figure 3 molecules-28-01663-f003:**
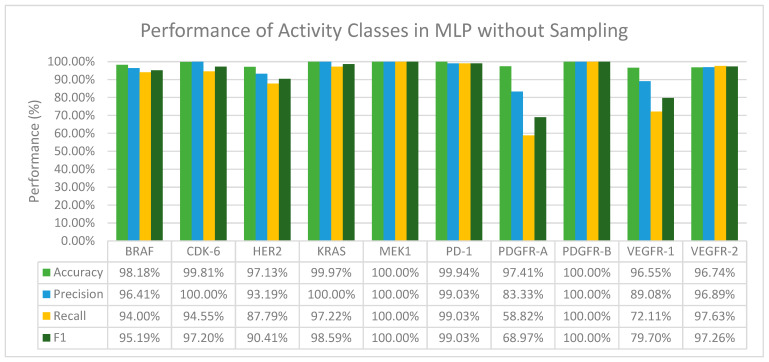
The performance of MLP in terms of accuracy, precision, recall and F1 score without sampling.

**Figure 4 molecules-28-01663-f004:**
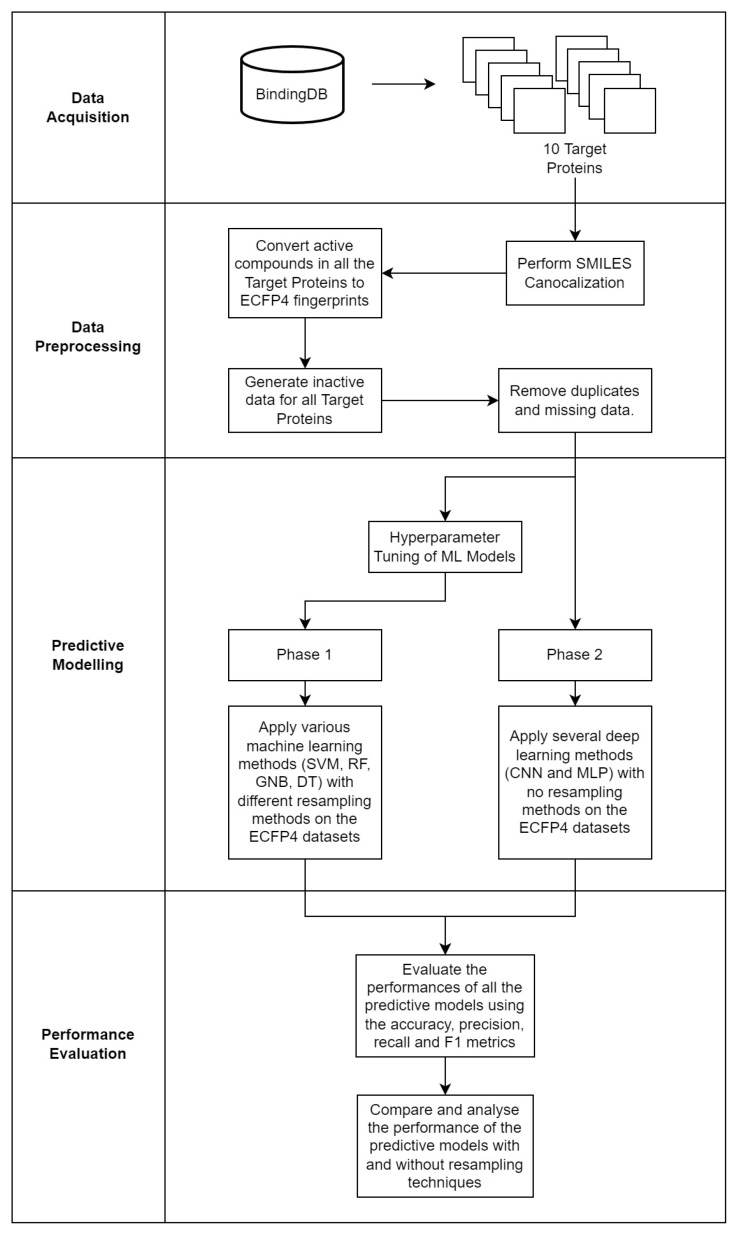
The four main phases of the study: data acquisition, data preprocessing, predictive modelling and performance evaluation.

**Figure 5 molecules-28-01663-f005:**
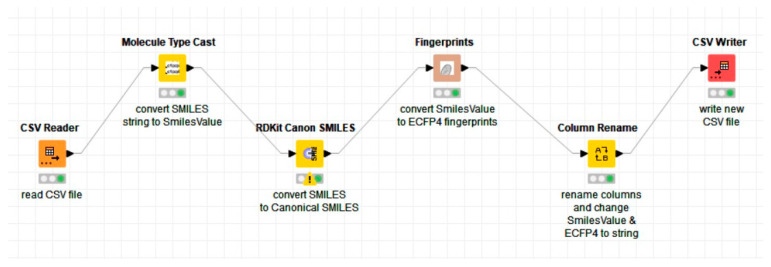
The workflow developed to convert the SMILES notation to canonical SMILES using the RDKit extension and then into chemical fingerprints (ECFP4) using the CDK extension in KNIME.

**Figure 6 molecules-28-01663-f006:**
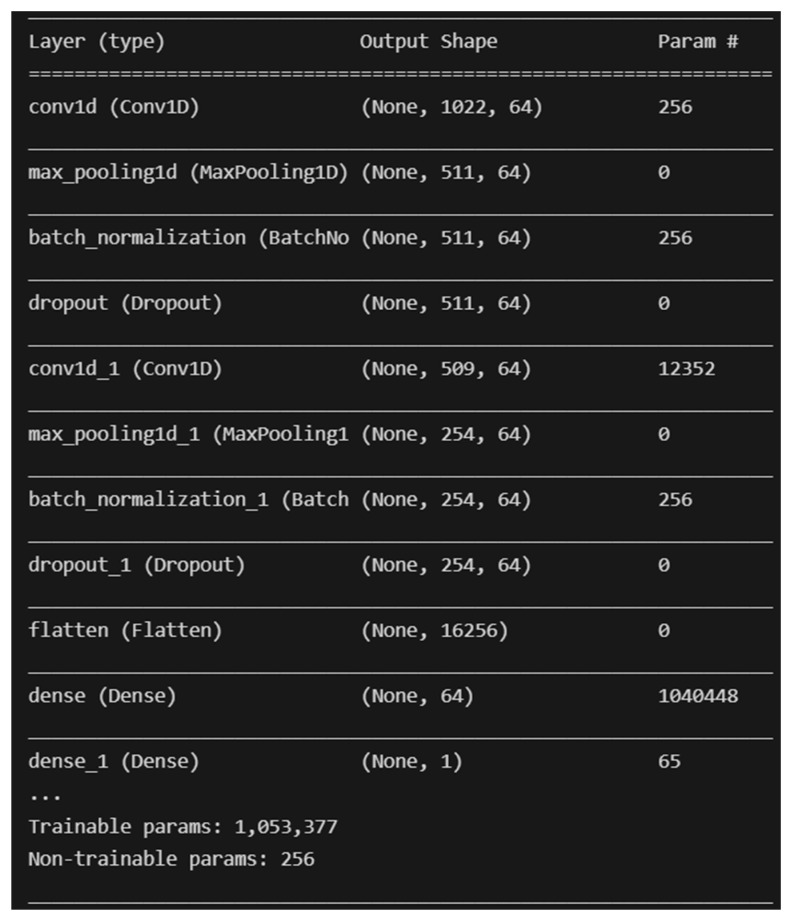
Overall architecture of the developed CNN model.

**Table 1 molecules-28-01663-t001:** The classification of the degree of imbalance for imbalanced data.

Degree of Data Imbalance	Proportion of Minority Class (%)
Severe	<1% of the dataset
Moderate	1–20% of the dataset
Mild	20–40% of the dataset

**Table 2 molecules-28-01663-t002:** The degree of imbalance and the percentage of minority class sorted by the number of active compounds per activity class.

Activity Classes	Number of Active Compounds	Percentage of Minority Class (%)	Degree of Data Imbalance
MEK1	36	0.23%	Severe
PDGFR-B	41	0.26%	Severe
KRAS	191	1.22%	Moderate
PD-1	470	3.00%	Moderate
CDK-6	581	3.71%	Moderate
PDGFR-A	724	4.62%	Moderate
VEGFR-1	1373	8.77%	Moderate
HER2	2426	15.50%	Moderate
BRAF	3071	19.62%	Moderate
VEGFR-2	9241	40.97%	Mild

**Table 3 molecules-28-01663-t003:** Best machine learning and resampling method pair for all cancer-related activity classes in terms of accuracy, precision and recall.

Activity Classes	Best Machine Learning and Resampling Method Pair		
	Accuracy	Precision	Recall
BRAF	RF + SMOTETomek	RF + None	GNB + SVM-SMOTE
CDK-6	RF + SVM-SMOTE	RF + None	GNB + SVM-SMOTE
HER2	RF + SMOTETomek	RF + None	GNB + SVM-SMOTE
KRAS	RF + SMOTETomek	RF + SMOTETomek	GNB + SMOTE
MEK1	RF + ADASYN	RF + ADASYN	GNB + SMOTETomek
PD-1	RF + ADAYSN	RF + ADAYSN	GNB + SVM-SMOTE
PDGFR-A	RF + SMOTE	RF + None	RF + BorderlineSMOTE
PDGFR-B	RF + BorderlineSMOTE	RF + BorderlineSMOTE	RF + RUS
VEGFR-1	RF + SMOTETomek	RF + None	RF + None
VEGFR-2	RF + SVM-SMOTE	RF + BorderlineSMOTE	RF + None

**Table 4 molecules-28-01663-t004:** Best machine learning classifier and resampling method pair for each activity with their respective F1 score.

Activity Class	ML + Resampling Pair	F1 Score
BRAF	RF + SMOTE	85.41%
CDK-6	RF + ADAYSN	96.00%
HER2	RF + SVM-SMOTE	89.71%
KRAS	RF + SVM-SMOTE	97.34%
MEK1	GNB + SVM-SMOTE	93.87%
PD-1	RF + ADASYN	99.36%
PDGFR-A	GNB + SVM-SMOTE	86.19%
PDGFR-B	GNB + SVM-SMOTE	85.68%
VEGFR-1	RF + None	93.50%
VEGFR-2	RF + BorderlineSMOTE	91.67%

**Table 5 molecules-28-01663-t005:** Comparison between the F1 score of machine learning paired with resampling and the F1 score of MLP.

Activity Class	Machine Learning with Resampling Pair	F1 Score of Machine Learning with Resampling Pair	F1 Score of MLP
BRAF	RF + SMOTE	85.41%	95.19%
CDK-6	RF + ADAYSN	96.00%	97.20%
HER2	RF + SVM-SMOTE	89.71%	90.41%
KRAS	RF + SVM-SMOTE	97.34%	98.59%
MEK1	GNB + SVM-SMOTE	93.87%	100.00%
PD-1	RF + ADASYN	99.36%	99.03%
PDGFR-A	GNB + SVM-SMOTE	86.19%	68.97%
PDGFR-B	GNB + SVM-SMOTE	85.68%	100.00%
VEGFR-1	RF + None	93.50%	79.70%
VEGFR-2	RF + BorderlineSMOTE	91.67%	97.26%
Average F1 Score		91.87%	92.36%

**Table 6 molecules-28-01663-t006:** Selected Activity Classes.

Target Proteins (Activity Classes)	Abbreviation	Number of Active Compounds
B-Raf Proto-Oncogene	BRAF	4389
Cyclin-Dependent Kinase 6	CDK-6	726
Human Epidermal Growth Factor Receptor 2	HER2	3046
Kirsten Rat Sarcoma 2 Viral Oncogene Homolog	KRAS	261
Dual Specificity Mitogen-Activated ProteinKinase Kinase 1	MEK1	38
Programmed Cell Death Protein 1	PD-1	479
Platelet-Derived Growth Factor Receptor A	PDGFR-A	864
Platelet-Derived Growth Factor Receptor B	PDGFR-B	41
Vascular Endothelial Growth Factor Receptor 1	VEGFR-1	1616
Vascular Endothelial Growth Factor Receptor 2	VEGFR-2	12,036

**Table 7 molecules-28-01663-t007:** The number of active and inactive compounds per activity class after data preprocessing and cleaning.

Target Protein (Activity Classes)	Abbreviation	Number of Active Compounds	Number of Inactive Compounds
B-Raf Proto-Oncogene	BRAF	3071	12,585
Cyclin-Dependent Kinase 6	CDK-6	581	15,075
Human Epidermal Growth Factor Receptor 2	HER2	2426	13,230
Kirsten Rat Sarcoma 2 Viral Oncogene Homolog	KRAS	191	15,465
Dual Specificity Mitogen-Activated Protein Kinase Kinase 1	MEK1	36	15,620
Programmed Cell Death Protein 1	PD-1	470	15,186
Platelet-Derived Growth Factor Receptor A	PDGFR-A	724	14,932
Platelet-Derived Growth Factor Receptor B	PDGFR-B	41	15,615
Vascular Endothelial Growth Factor Receptor 1	VEGFR-1	1373	14,283
Vascular Endothelial Growth Factor Receptor 2	VEGFR-2	9241	6415

**Table 8 molecules-28-01663-t008:** Brief descriptions of the resampling methods used in this study.

Resampling Method	Brief Description of Resampling Method	Authors & Year
SMOTE	Synthetic Minority Oversampling Technique or SMOTE was introduced by the authors in [54]. SMOTE is an oversampling technique that oversamples the minority class in the dataset by synthesizing fake minority data into the original dataset so that the minority and majority classes are balanced.	Chawla et al. (2022) [54]
RUS	Random Undersampling or RUS is an undersampling technique that randomly discards instances from the majority class so that the majority class is balanced with the minority classes [55].	Lemaitre et al. (2016) [55]
ADASYN	ADASYN or Adaptive Synthetic, introduced by the authors in [56] is a data resampling technique that is similar to SMOTE, but in ADASYN, synthetic data are generated for minority classes that are difficult to learn.	Haibo et al. (2008) [56]
BorderlineSMOTE	BorderlineSMOTE is a resampling method that was developed by the authors in [57]. BorderlineSMOTE is an oversampling method where only minority samples that are near the borderline are oversampled.	Han et al. (2005) [57]
SVM-SMOTE	SVM-SMOTE is a resampling method that was developed by [58] that is similar to BorderlineSMOTE. The difference is that, in this method, SVM is used to help generate new minority instances near the borderline in order to establish boundaries between the classes in the dataset.	Nguyen et al. (2011) [58]
SMOTETomek	SMOTE with Tomek Links or SMOTETomek is a hybrid resampling technique that combines oversampling using SMOTE and undersampling using Tomek links and it was introduced by [59].	Batista et al. (2003) [59]

**Table 9 molecules-28-01663-t009:** Machine learning classifiers with their optimized parameters after hyperparameter tuning.

Machine Learning Classifier	Best Parameters
SVM	C = 100, gamma = 0.1, kernel = sigmoid
RF	criterion = entropy, max_depth = 8, min_samples_leaf = 2, min_samples_split = 5, n_estimators = 100
GNB	var_smoothing = 0.1
DT	ccp_alpha = 0.001, criterion = entropy, max_depth = 4, max_features = auto, min_samples_leaf = 5

## Data Availability

The data presented in this study are available on request from the corresponding author.

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
