# Peer review of "Comparative Studies on Resampling Techniques in Machine Learning and Deep Learning Models for Drug-Target Interaction Prediction"

_molecules, 2023, doi:10.3390/molecules28041663_

Round 1
Reviewer 1 Report
The article aims to evaluate the suitability and performance of several machine learning methods (with or without using resampling techniques to overcome class imbalance) in classifying imbalanced datasets. To this end, the authors have performed extensive computational experiments based on the real-word activity data for several anti-cancer targets. The results are quite interesting and will likely be useful to QSAR and chemoinformatics practitioners. The article can be published after a MINOR revision addressing mostly the presentation issues.
1) The detailed explanation of the process whereby “the inactive data for each of the activity class is generated through a dataset generation program”. In particular, where do the inactive (or supposedly inactive) structures come from?
2) The hyperparameter tuning procedure should be clarified. It is not clear how “Hyperparameter tuning is done so that the model with best parameters is determined before feeding any data into it for learning”, as tuning without data is meaningless and the goal is actually to determine optimal parameters for a specific problem or class of problems (otherwise we could have been able to find the “universally best” set of parameters).
3) The architectures and hyperparameters used for the CNN and MLP neural network models (as well as the considerations or procedures that were used to determine them) should also be specified in detail. E.g., the set of layers, number of neurons, CNN kernels, activation functions, training methods, number of epochs, etc.
4) It is not clear why SMILES canonicalization was necessary before the fingerprint calculation. Being molecular graph invariants, the fingerprints should not depend on particular SMILES representations (that all should yield the same molecular graph). In addition, why the CDK fingerprints were preferred over the fingerprints available in RDkit?
5) The Results and Discussion section devotes several pages to tedious narration of detailed results obtained in each case, that are further duplicated in many pages of plots and tables. Most of these data should be moved to the Supplementary Materials, providing instead a brief and easily readable explanation and analysis of key findings. It should also be noted that the classification accuracy is well known to provide misleading results for imbalanced datasets (in fact, that is the main reason behind the introduction of precision, sensitivity/recall, specificity, and many other performance metrics).
6) General style of the article should be improved. At times, it reads like a student report with lots of excessive details and unnecessary repetitions. In most cases, listing the author names of the cited publications should be avoided (sometimes, coupled with incorrectly formatted references, it also leads to strange phrases like “according to Google Developers”). The dichotomy of “natural products or chemical compounds” is misleading, as the natural products (at least those relevant to medicinal chemistry) ARE the chemical compounds. Proteins are not usually represented as SMILES notation. The assertion that “deep learning is the most successful technique in drug discovery” is somewhat over-sweeping. In addition, deep learning is not unique in supporting multi-task learning – non-deep neural networks and many other machine learning approaches also can work with multiple classes. The format of Tables 3 and 4 should be corrected. English in the article is mostly good but should be tidied up a bit in some places with respect to grammar, terminology, and style (e.g., “pharmacists”, among others).
Author Response
Dear Reviewer,
Please refer to the attachment.

Author Response
Dear Reviewer,
Please refer to our attachment.

Round 2
Reviewer 2 Report
Author's have successfully addressed all comments.
Manuscript is now suitable for publication.